# COVID-19 Infection Detection and Prevention by SARS-CoV-2 Active Antigens: A Synthetic Vaccine Approach

**DOI:** 10.3390/vaccines8040692

**Published:** 2020-11-18

**Authors:** José Manuel Lozano, Luz Mary Salazar, Ángela Torres, Adriana Arévalo-Jamaica, Carlos Franco-Muñoz, Marcela Mercado-Reyes, Fabio Ancizar Aristizabal

**Affiliations:** 1Departamento de Farmacia, Universidad Nacional de Colombia- Sede Bogotá, Carrera 30#45-03, Bogotá DC 111321, Colombia; faaristizabalg@unal.edu.co; 2Grupo de Investigación: Mimetismo Molecular de los Agentes Infecciosos, Departamento de Farmacia, Universidad Nacional de Colombia-Sede Bogotá, Carrera 30 #45-03, Bogotá DC 111321, Colombia; lmsalazarpu@unal.edu.co (L.M.S.); agtorresr@unal.edu.co (Á.T.); 3Departamento de Química, Universidad Nacional de Colombia- Sede Bogotá, Carrera 30 #45-03, Bogotá DC 111321, Colombia; 4Dirección de Investigación en Salud Pública, Instituto Nacional de Salud, Bogotá 111321, Colombia; aarevalo@ins.gov.co (A.A.-J.); cfranco@ins.gov.co (C.F.-M.); mmercado@ins.gov.co (M.M.-R.); 5Instituto de Biotecnología (IBUN), Universidad Nacional de Colombia-Sede Bogotá, Carrera 30 #45-03, Bogotá DC 111321, Colombia

**Keywords:** SARS-CoV-2, COVID-19, antigen design, synthetic-vaccine candidate

## Abstract

COVID-19, a global pandemic causing to date more than 50 million cases and more than a million deaths, has to be controlled. SARS-CoV-2 (severe acute respiratory syndrome coronavirus 2) was identified as the causative agent. Controversy about this virus origin and infectious mechanism for adapting to humans remains a matter for discussion. Among all strategies for obtaining safe and potent vaccines, approaches based on attenuated-killed virus and non-replicating RNA viral vectors are demonstrating promising results. However, specificity of viral components targeted by human antibodies so far has not been demonstrated. A consistent strategy for obtaining functional-active antigens from SARS-CoV-2 specific ligands lead us to propose and test a number of synthetic components. From hundreds of starting sequences only fifteen fulfilled the design requirements and were produced as monomer and polymer forms and immuno-chemically tested. The design was based on worldwide representative reported virus genomes. A bioinformatics scheme by conventional methods and knowledge on MHC-I and II antigen processing mechanisms and HLA haplotype-restriction was performed including sensitive and resistant human populations to virus infection. Covid-19 patients’ sera reactivity for synthetic SARS-CoV-2-designed components have proven a high recognition of specific molecules, as well as some evidence for a long-lasting humoral immune response.

## 1. Introduction

COVID-19, also known as new coronavirus disease, is a severe respiratory acute syndrome in humans caused by the positive strand RNA SARS-CoV-2 virus which belongs to the beta-coronavirus family (β-CoVs) and which became a severe epidemic, currently having taken more than a million lives and infected close to fifty one million people worldwide between December 2019 and November 2020 [1]. The family of CoVs is a class of enveloped positive-sense single-stranded RNA viruses having an extensive range of natural origins. These viruses can cause respiratory, enteric, hepatic, and neurologic diseases.

The SARS-CoV-2 is composed of a 30 kb genome coding for sixteen non-structural proteins (NSPs), four structural proteins (SPs) named S, E, M and N, as well as some accessory proteins (APs) under control of ten open reading frames (ORFs). Genome organization of SARS-CoV-2 is known as 5′-leader-UTR-replicase-S(Spike)-E(Envelope)-M(Membrane)-N(Nucleocapsid)-3′-UTR-poly(A) tail. Accessory genes are interspersed within the structural genes at the 3′-end of this genome. The first Wuhan (people’s republic of China) SARS-CoV-2 virus isolated was reported and stored under the National Center of Biotechnology Information (NCBI), U.S. National Library of Medicine, as the reference sequence coded NC_045512.2. Then, worldwide virus sequence isolates were reported including Brazil (EPI_ISL_412964 and EPI_ISL_412964), Italy (MT066156.1), and Colombia (EPI_ISL_417924) as being the eighth most infected nation in agreement with data from more than 180 countries. Population genetic evaluations of more than 100 SARS-CoV-2 genome sequences, shown that this virus family possess two major lineages known as S and L, the L being more prevalent than the S lineage.

Among structural proteins, a capsid containing the positive single strand (ss) viral RNA anchor a protein termed spike (S) coded by the ORF2 which is a 1288 amino-acids protein of apparent molecular weight of 142 kDa (GeneID: 43740568) [2]. Spike is composed of two S1 (700 amino-acids) and S2 (600 amino-acids) sub-units. S1 includes the receptor binding domain (RBD) from 333 to 527 position, which binds its human receptor to the angiotensin converting enzyme 2 (ACE2), a protein which is ubiquitously present in almost all human cells, tissues and organs including lungs, hearth, liver and kidneys. Three S1/S2 hetero dimers are assembled to form a trimer spike protruding from the viral envelope. The spike trimeric molecular arrangement on each single chain contains an ectodomain composed of sites 1 and 2 responsible for receptor binding and cell entry having a Kd of ~15 nM [3,4]. Interestingly, a highly conserved cryptic epitope on the spike RBD was identified [5]. Recently, evidence demonstrating that SARS-CoV-2 cell entry through ACE2 can be inhibited by specific compounds was reported [6].

It is known that the virus unit ranges between 60~100 nm of apparent diameter and appears round shaped. A simple scheme for genome and structure organization for the SARS-CoV-2 can be observed in Figure 1a. The virus harbors 10 open reading frames (ORFs) coding for all structural, non-structural, and accessory proteins as described in Figure 1b.

The S protein of SARS-CoV-2 belongs to a transmembrane glycoprotein family having a predicted size of 1255 amino-acids possessing a leader sequence from residues 1 to 14, an ectodomain from residues 15 to 1190, a transmembrane domain from 1191 to 1227 residue, and a short intracellular tail from 1227 to 1255 residue, as described elsewhere. Interestingly, the trimeric spike structure conformation possesses both a closed and an open form as recently published [7]. Structure coordinate files for both the open and closed spike protein-states were stored under codes 6vsb and 6vxx, respectively, in the protein data bank (PDB). The spike RBD- ACEII 3D structure-complex was deeply studied and filed with the 6m0j PDB code [8,9].

Likewise, an envelope E gene is coded by the ORF4. E protein is a small polypeptide (76–109 amino acids) of apparently 10 kDa molecular weight and a pI of 8.57 and contains a single alpha-helical transmembrane domain. It is arranged as a pentamer protein on the virus capsid surface and has been identified as YP_009724392.1 (GeneID: 43740570) and its 3D structure coordinates were stored (PDB code 5x29) [9]. The E gene expression medium size polypeptide product is known as a non-glycosylated small transmembrane protein, and it appears to act as a molecular engine promoting the SARS-CoV-2 assembling in the host cytosolic compartments such as Golgi complex and the endoplasmic reticulum [10].

A virus matrix named M protein, is coded by the ORF5, and identified as YP_009724393.1 (GeneID: 43740571), is a glycoprotein of 25–30 kDa and is highly abundant on the virus surface. It is known that M interacts with E protein, and so it seems to be relevant for the SARS-CoV-2 maturation, as such M becomes a key piece for the virus assembling. To date, the M protein 3D structure properties have not been reported to the PDB.

On the other hand, a nucleocapsid phospho-protein known as N is coded by the ORF9 (GeneID: 43740575). This gene’s expression product has a molecular weight between 45 and 50 kDa (YP_009724397.2). It is known that N is the most conserved among all structure proteins of coronaviruses, it appears to be required in virus RNA encapsidation, and it seems to be relevant for the virus replication. Protein dimers of N are assembled in hexamers and such complexes’ 3D structure coordinates have been stored under PDB codes 6m3m and 6 wkp, respectively [9].

The membrane proteins S, E and M are inserted into the intermediate compartment of the virus capsid while the viral RNA undergoes replication as being assembled in the N protein. This RNA-protein complex is associated with the endoplasmic reticulum membrane-inserted M protein, allowing the virus to assemble and migrate to the Golgi complex and an eventual virus release from the host cell can occur by exocytosis.

Additionally, 16 non-structural proteins (NSPs) are coded by ORF1a/1b and actively participate in the virus RNA replication. Some accessory proteins of non-well understood functions are coded by genes from ORF3a, 3b, 6, 7a, 7b, 8, 9b, 9c and 10. All SARS-Cov-2 protein 3D structures were revised from their primary sources [11].

On the other hand, a global effort started with more than 140 vaccine candidate proposals for a SARS-CoV-2 vaccine but only two have been currently approved despite lacking safety information thereof. The most promising 37 vaccine candidates were focused on approaches based on non-replicating RNA virus vectors and inactivated virus, and a few of them are viral inserts on double-stranded foreign DNA, and others presented as recombinant protein sub-units, most of them based on the spike protein structure [12]. To date, most of these vaccine candidates are enrolled on Phases I/II/III clinical trials evidencing encouraging results. However, these candidates’ safety and potency become a real concern for the scientific community [13,14]. Due to the importance of having vaccine candidates able to be used in large human populations, synthetic strategies emerged as an alternative pathway towards specific, safe and efficient vaccine candidates. In addition, animal models and ex vivo SARS-CoV-2 virus neutralization tests for assaying COVID-19 vaccine candidate stimulated antibodies are relevant matters which are undergoing testing world-wide.

In the present work, we present evidence of antigenic activity of synthetic site-directed designed components based on SARS-CoV-2-structure when faced with the antibodies of patients with COVID-19, revealing the potential of an envisioned strategy towards infection detection, and virus vaccine candidate selection. Evidence of long-lasting antibody immunity is briefly discussed.

## 2. Materials and Methods

### 2.1. Virus Genome, Amino-Acid Sequence Analyses and Target Epitopes Designing

Worldwide, SARS-CoV-2 reported genomes were the basis for designing and obtaining the main amino-acid sequences and peptides reported in the present work. Therefore, genome data to isolate the SARS-CoV-2 virus and sequencing information was downloaded from NCBI and GISAID databases [11,15].

A second step, consisting of a multiple sequence alignment of mostly SARS-CoV-2 reported genomes was performed using the Clustal omega tool of EMBL-EBI, Wellcome Genome Campus, Hinxton, Cambridgeshire, UK and led to the establishment of a ≥ 96% identity value among all compared sequences. Hence, a Basic Local Alignment Search Tool (BLAST) from the NCBI National Center for Biotechnology Information, U.S.A, led to the performance of sequence analyses for all ORFs coding for structural, non-structural and accessory SARS-CoV-2 proteins.

Subsequent in silico analyses consisted of submitting each ORF coding sequence to identify the presence of possible LB epitopes, including the presence of proteasome cleavage sequence sites, HLA-I and HLA-II different length binding epitope sequences regarding endosomal and phagosome-lysosome protease cleavage sites, by accessing remote servers for B-cell epitope prediction, also known as B lymphocyte epitopes as linear arrangements thereof or Linear B (LB) epitope prediction with LBtope and ABCpred main page bearing a threshold: 0.51. Other bioinformatics tools used were used for B-cell epitope prediction [16,17,18].

For predicting proteasome cleavage, the IEDB server was employed [19]. HLA-I binding motifs were analyzed with netMHCpan v4.0 and NetMHCpan-2.3 servers. HLA-II binding motifs from SARS-CoV-2 were analyzed with the netMHCpan v3.2 and a IEDB tool were also employed. To identify phagosome-lysosome protease cleavage sites the PROSPER server was used [20,21,22,23,24].

For obtaining target sequences to be synthesized, more than 100 amino-acid sequences in a range from 10 to 23 residues in length passed the first design filter, and some selection criteria were then considered, including those potential epitopes showing binding scores from 50 to 100 nM. Those sequences which had simultaneously identified both proteasome cleavage sites, LB epitopes and HLA-I binding sequences were pre-selected. Among preliminary sequences from this filter, those matching HLA-II binding sequences, proteasome including potential phagosome-lysosome cleavage sites, were regarded for a further selection step. A molecular map consisting of all the preliminary sequences from each SARS-CoV-2 consensus ORFs was built and those coincidences among susceptible and resistant infection HLA-I and HLA-II were assessed as inclusion criteria to be considered in the selection list. Considerations regarding HLA haplotype global distribution and their impact on resistance and sensitivity to SARS-CoV-2 infection led us to define the candidate proposal among all sequence candidates. The non-polymorphic-screened unique sequences were regarded as the source of a final preliminary antigen list consisting of 15 different epitope-peptides to be synthesized by solid phase peptide synthesis by standard Fmoc (9-methyl fluorenylmethoxycarbonyl) procedures based on literature reported elsewhere [25].

Representative peptides from structural and accessory ORFs-expressed proteins of SARS-CoV-2 were then modified by some amino-acid substitutions at given residue positions. Thus, different length peptides ranging from 10 to 23 residues from ORFs 2, 3a, 3b, 4, 5, 7a, 7b and 9, including accessory and structural S, E, M and N proteins, were selected as the target sequences of this work. Therefore, peptide sequences presented as single monomer and polymer forms were designed. Monomer peptide sequences are represented by the single-viral-motif (SVM) code beside an odd number and their polymer peptide forms are represented by the polymer-hybrid-element (PHE) code besides a pair number. Thus, a single sequence will be named in agreement with its molecular state, the monomer having an odd number and its polymer form with a consecutive even number.

### 2.2. Synthesis of Monomer and Polymer Forms for Each Epitope-Sequence

Fifteen epitope-sequences were obtained by solid phase synthesis by 9-fluorenylmethoxycarbonyl (Fmoc) strategy as monomer and polymers form for a total of 30 polypeptides. Solvents and soluble reagents were removed by filtration. Washings between deprotection, couplings and subsequent deprotection steps were carried out with *N,N*′-dimethylformamide (DMF) (5 × 1 min), dichloromethane (DCM) (4 × 1 min), isopropyl alcohol (IPA) (2 × 1 min) and DCM (2 × 1 min) using 1.5 mL of solvent/50 mg of resin each time. The Fmoc group was removed from the resin by two treatments of 15 min with piperidine-DMF (25:75 *v*/*v*). Couplings were performed at 20 °C and monitored using standard Kaiser tests for solid-phase synthesis.

For the synthesis of monomer forms, after Fmoc removal of the commercially available Rink amide resin (50 mg, 0.46 mmol/g), the first Fmoc-amino-acid (0.115 mmol, 5.0 equiv.) was added with 1-hydroxybenzotriazole (HOBt) (18.2 mg, 0.115 mmol; 5.0 equiv.) and N,N′-dicyclohexylcarbodiimide (DCC) (23.7 mg; 5.0 equiv.) as coupling reagents dissolved in DMF/DCM (7:3, *v*/*v*) and the coupling reaction was stirred for 2 h. Next, the Fmoc group was removed, and a second Fmoc-amino-acid was incorporated in the resin using the same conditions. The Fmoc removal and the coupling reactions of the rest of the Fmoc-amino-acids were carried out under the same conditions using 5 equiv./coupling. Finally, monomer peptide was Fmoc deprotected and cleaved from the resin by treatment with a mixture of trifluoroacetic acid- water- triisopropylsilane (TFA/H_2_O/TIS) (95.0:2.5:2.5) for 6 h followed by filtration and precipitation with cold diethyl ether (Et_2_O). Crude products were then triturated 3 times with cold Et_2_O, dissolved in the system water-acetonitrile (H_2_O:CH_3_CN) (9:1 *v*/*v*) and then lyophilized.

Synthesis of polymer forms was carried out on 150 mg of Rink-amide resin. In order to further obtain a molecule of high molecular weight represented by a polymer, an active cysteine residue was incorporated at both *N-* and *C-* sequence ends. The synthesis of polymer forms was carried out under the same strategy and conditions used for their corresponding monomers (5 equiv./coupling).

Synthesized Cys-peptides were Fmoc deprotected and cleaved from the resin by treatment with a cleavage mixture including ethanedithiol (EDT) in the system TFA/H_2_O/TIS/EDT (94.0:2.5:1.0:2.5) for 6 h followed by filtration and precipitation with cold Et_2_O. These crude products were then triturated 3 times with cold Et_2_O and dissolved in H_2_O:CH_3_CN (9:1 *v*/*v*) and lyophilized as before. Finally, cysteinyl peptides were submitted to disulfide bridge oxidation to obtain the target polymeric molecular forms. Oxidation was carried out from a peptide solution in water (4 mg/mL, pH 7.0) by an oxygen stream under stirring for 16 h. Polymer peptides obtained were dialyzed in water for 24 h using a 500 Da cellulose acetate membrane and further lyophilized as previously published [26]. Monomer and polymer SARS-CoV-2 sequences were employed for serological tests with human sera samples, as well as for further biological assays.

### 2.3. Peptide Characterization

All SARS-CoV-2-based peptide antigen monomer and polymer forms were characterized by analytical reverse phase-high performance liquid chromatograph (RP-HPLC) and analyzed by matrix-assisted laser desorption/ionization- time-of-flight (MALDI-TOF) mass spectrometry. Analytical RP-HPLC was performed using an Agilent 1200 series chromatography system (Agilent Technologies, Inc., Santa Clara, CA, USA). Analyses were performed on a Zorbax^®^ HPLC C18 column (4.6  ×  50 mm, 5 µm) (Merck KGaA, Darmstadt, Germany), 1 mL/min rate flow, mobile phase system was A: H_2_O/TFA (99.9:0.1 *v*/*v*); and B: CH_3_CN/TFA (99.9:0.06 *v*/*v*), on a 5% to 95% of B linear gradient during 5 min at a 25 °C temperature and the UV detector was adjusted to 220 nm. A MALDI-TOF mass spectrometry was carried out to the external service to identify a molecular ion of each peptide. Fmoc-Rink Amide MBHA resin and Fmoc-amino-acids were purchased from Iris Biotech GmbH (Marktredwitz, Germany); DCC and HOBt from AAPPTec (Louisville, KY, USA); piperidine, EDT, TIS and TFA were purchased from Sigma–Aldrich (Steinheim, Germany) and Et_2_O, DMF, DCM, IPA, CH_3_CN, from Merck KGaA (Darmstadt, Germany). All commercially available reagents and solvents were used as received without further purification. Distilled and deionized water was used for the preparation of all solutions and chromatography eluents.

### 2.4. Serological Test and Statistical Analysis

ELISA standard assays were performed for the detection of antibodies to SARS-CoV-2 from sera samples of humans with COVID-19. Polystyrene 96-flat bottom plates (Thermo Fisher Scientific, Waltham, MA, USA) were immobilized overnight with representative amounts of each synthetic antigen ranging from 5 µg/mL to 40 µg/mL in a carbonates/bicarbonate buffer at a pH of 9.6 at 4 °C. Following washings and non-specific binding blocking steps with a solution of 1–5% of skimmed-milk in 0.15 M (phosphate buffered solution) PBS- 0.05% (*v*/*v*) tween-20, different dilutions of human sera samples were poured onto peptide-ELISA plate wells and incubated for 1 to 3 h at 37 °C. Then a goat alkaline phosphatase anti human-Ig-conjugate was poured at different dilutions on PBS-Tween-20 and incubated for one-hour at 37 °C to bind specific human antibodies to SARS-CoV-2 epitopes. After performing standard washings, the test was developed with a 1 mg/mL p-nitrophenylphosphate solution in 0.1 M diethanolamine, pH 9.8 to reveal those antibodies’ specific binding to designed virus epitopes by a yellow color appearance which was then detected on a microplate-reader (Multiscan EX^®^, Thermo Fisher Scientific, Waltham, MA, USA) adjusted at 405 nm. When necessary, enzyme activity was stopped by adding a 3 N NaOH solution before absorbance reading. Human sera samples were collected under Colombian and international ethical regulations (Presidency of the Republic of Colombia, decree number 266, 2006), in agreement to the world medical association WMA- Declaration of Helsinki of ethical principles for medical research involving human subjects, from either symptomatic or asymptomatic patients with COVID-19, were tested by a standard PCR for a positive viral charge and informed previous blood sampling as reported [27]. A follow-up for serum antibody conversion was accorded with patients involved in the present research. GraphPad Prism 7 software was used for statistical analyses.

### 2.5. SARS-CoV-2 Protein Structure Analysis

Information of the SARS-CoV-2 ORF gen coding for those 3D structure resolved proteins was obtained [11], and PDB coordinate files were downloaded from the protein data bank site (PDB) [10] as mentioned above. Protein molecular modeling and personalization was performed using downloaded available protein coordinate files whose PDD code files were, for S (open and close states 6vsb and 6vxx), S-RBD-ACE II (6m0j), E (5 × 29), M (not yet reported) and N (6m3m and 6 wkp), respectively. For those proteins whose 3D structure has not been reported yet to the PDB, molecular modeling was carried out by submitting epitope-sequences to remote servers to obtain predicted structure- homology models in coordinate files, being the PepFold structure-prediction server selected to fulfill this aim [28,29]. Thus, PDB files for representative epitope-peptides and their modified versions for ORF3a, ORF5 (M protein), ORF7a and ORF7b were achieved, all fulfilling the Ramachandran plot structure requirements, as well as the geometrical constraints and restraints parameters for a valid protein structure. Finally, personalized molecular modeling for displaying all PDB 3D structure coordinate files, were allowed by using the VMD 1.9.3 version software, from the NIH Biomedical Research Center for Macromolecular Modeling and Bioinformatics, University of Illinois [30].

## 3. Results

### 3.1. Epitope-Peptide Design and Structure Characterization

A molecular design to determine the most likely epitope-peptides from SARS-CoV-2 virus was based on concerted strategies in which was used not only bioinformatics basin, but experience and knowledge on peptide chemistry, vaccinology and infectious diseases transmission. Therefore, all ten of the virus’ OFRs coding for a number of proteins among structural, accessory and non-structural were submitted to a deep analysis and theoretical characterization. Screening of potential epitopes derived from the virus’ most representative proteins lead us to propose fifteen amino-acid sequences out of more than one hundred that were preselected, having different sizes ranging from 10 to 23 residues in length and different hydrophobic profiles. These selected 15 peptide sequences were synthesized by Fmoc chemistry showing average yields ranging from 65% to 85% and were produced as monomer (SVM) and polymer (PHE) forms, under controlled production steps following standard protocols for GMP production and analysis. Purification lead to obtaining purities higher than 90% for most molecules. Representative polymerized epitope-peptides were obtained and termed PHE as mentioned, thus from proteins coded by ORF2 (spike S) peptides were PHE-2, PHE-4, PHE-6, PHE-8 PHE-10, PHE-12 and PHE-14, from ORF3a PHE-16; for ORF3b PHE-18; for ORF4 (envelope E protein), PHE-20; for ORF5 (M surface protein) PHE-22; for ORF7a PHE-24; for ORF7b PHE-26 and for ORF9 (Nucleocapsid protein) two sequences coded PHE-28 and PHE-30, respectively. The relative position of each epitope-peptide on its parent SARS-CoV-2 protein 3D structure can be observed in Figure 2.

### 3.2. Serological Reaction to Synthetic Epitope-Peptides of Antibody-Serum from Individuals being RAS-CoV-2 Positive and Negative PCR Tested

We decided to assess each single obtained epitope-peptide representative of target SARS-CoV-2 proteins as monomer and polymer form thereof. Therefore, as observed in Figure 3a, the sera reactivity of PCR-diagnosed patients recognizes in higher OD magnitudes to those epitope-peptides presented as polymer forms, as expected, that the multi-copy presented epitopes would give the chance of being recognized by human antibodies from patients with SARS-CoV-2 as seen for PHE-4, PHE-6, PHE-10, PHE-12 and PHE-14 all representing different epitopes of the spike protein (coded by ORF2) at different sequence positions including the RBD to ACE2. With the aim of testing a possible epitope synergy-effect for antibody recognition, monomer and polymer forms thereof were pooled in an equimolar ratio and subsequently tested for their immuno-reactivity.

As observed in Figure 3b, optical densities drastically enhanced their magnitude, especially those regarding recognition of epitope-polymer forms as previously evidenced, this time faced with 4 positive and 2 negative sera samples. Subsequently, epitope-peptides representative of ORFs 3a, 3b, 4, 5, 7a, 7b and 9 were also tested for immuno-reactivity against a number of human PCR positive and negative sera samples as both monomer and polymer forms. As shown in Figure 3c, polymer-presented epitope-forms seem to be better recognized by all positive diagnosed patients in a differential fashion. The controls 2 sequences from the influenza A H1N1 virus (codes C31 and C 33) were tested simultaneously as observed. Thus, polymer epitope-peptides PHE-20 (E protein), PHE-22 (M protein), PHAE-24 (ORF7a) and PHE-28 (ORF9, N protein) were highly recognized beside some monomer forms thereof, such as SVM-19 (E protein), SVM-21 (M protein) and SVM-27 (N protein).

We then performed ELISA tests to assess a possible serological reaction of COVID-19 positive (*n* = 20) and negative (*n* = 20) serum samples using synthetic epitope-peptides as the antigen (Figure 4). It is important to bear in mind that the design of the epitope-peptides considered within the strategy the genomic information of SARS-CoV-2 lineages present in Colombia and targeting several structural proteins of SARS-CoV-2 including spike, nucleocapsid, envelope and others as mentioned above. Most of the epitope-peptides showed higher absorbance values when faced with serum from COVID-19 positive diagnosed individuals than faced with that of the negative diagnosed ones. The mean differences in absorbance were significant (*p* < 0.05) for epitope-peptides PHE-6 (Figure 4a), PHE-28 (Figure 4g) and PHE-30 (Figure 4h).

A five-month follow-up study of Covid-19 symptomatic patients over after being PCR diagnosed by assaying the whole set of epitope-peptides representatives from structural and accessory proteins of SARS-CoV-2. Therefore, serum samples obtained after one and five months of PCR diagnosis were challenged against the synthetic epitope-peptides by a standardized ELISA test (Figure 5).

Interestingly, having obtained the results evidenced after one month of being challenged by the virus, all synthetic epitope-peptides were more immuno-reactive for specific components when faced to the patient serum compared with historical pre-pandemic sera obtained for a representative number of human beings. Remarkably, as showed in our experiments, after five months of the virus infection, some epitope-peptides have an increase in their immuno-reactive capacity revealing a strong humoral immune response to selecting epitopes, suggesting a long-lasting humoral immunity caused by the virus itself in controversy to recently published data [31].

However, reactivity to some other epitope-peptides significantly reduces their immuno-reaction by patients’ sera to levels near to those revealed by negative sera. Hence, these results could help to explain the performance of some IgG-detection based tests, which mostly fail in the screening of patients exposed to SARS-CoV-2 in COVID-19 seroprevalence studies, and this fact could be related to a short-lasting immunity for some specific SARS-CoV-2 antigens. Nevertheless, it is important to point out that data presented herein corresponds to a case study, among a number of observations of many patients, which reveals that the epitope-peptides PHE-6, PHE-14 from the spike protein, PHE-20 from the envelope E protein and PHE-28 and PHE-30 from the nucleocapsid protein seem to be the higher immuno-SARS-CoV-2 reactive targets, while immuno-reactivity for PHE-22 and PHE-24 sequences from M and ORF7a decreases over the observation time.

## 4. Discussion

A molecular design aimed to propose some epitope-peptides was conducted considering multiple factors associated with the SARS-CoV-2 virus that is most representative world-wide reported genomes for obtaining a representative virus set of epitopes. Molecular designing steps were built to fulfill hypotheses regarding a proper selection including reasoned on antigen presentation in the classes I and II contexts, as well as proteasome and phagosome-lysosome cleavage preferences and frequencies bearing world populations sensitive and resistant to the virus infection. Bioinformatics tools were useful for this aim, but knowledge and experience in immunogenic molecules design were key pieces for this complex molecular puzzle fixing, which is still in progress beyond this work. A number of sequences higher than one hundred were regarded as key epitope-peptides representative of S, E, M and N proteins as well as expression products from ORFs 3a, 7a, lead to identify 15 target sequences which were then obtained and characterized as being monomer forms herein identified as single-viral-motifs (SVMs), as well as their polymerized forms denoted as polymer-hybrid-elements (PHEs). Interestingly, PHEs were more highly recognized by human antibodies than their monomer counterparts, probably due to the high conformer number of a given sequence presented as a polymer.

Serum samples from COVID-19 diagnosed Colombian patients, assessed by the standard SARS-CoV-2 PCR- based methodology, among those asymptomatic and symptomatic at different clinical conditions, were assessed in seroprevalence studies regarding those selected SARS-CoV-2 epitope-peptides. As observed, antibodies from sera samples were able to differentially recognize the most representative designed epitope-peptides, evidencing in some particular cases a long-lasting immune response to specific epitopes from N, S and E, while the opposite effect can be seen for epitope-peptides from M and ORF7a, this being a controversial matter that should be assessed in further assays, as well as specific B cell clones’ stimulation by given epitopes. Experiments to evaluate antibody-stimulation under controlled vaccination schemes, safety and human adjuvant system formulations of the herein reported epitope-peptides on animal models are undergoing processes in our research group. Increasing the number of individuals from different world regions, ethnicity, gender, age and different clinical status should be further considering as an important step towards finding universal answers to the current COVID-19 pandemic.

## 5. Conclusions

Altogether, the obtained results encourage us to propose some of the herein presented molecules as possible tools to be considered at different levels and conditions for the detection and prevention of patients with SARS-CoV-2. Therefore, a vaccine formulation composed of site-directed designed synthetic epitope-peptides constitutes an attractive approach and a reliable conceptual contribution for preventing COVID-19.

## 6. Patents

Provisional patent applications for designed epitope-peptide amino-acid sequences were filed under EFS IDs 40568238 and 40943743, whose application numbers were 63078840 and 63105478, respectively, on 15th September 2020, in the United States Patent and Trademark Office-USPTO.

## Figures and Tables

**Figure 1 vaccines-08-00692-f001:**
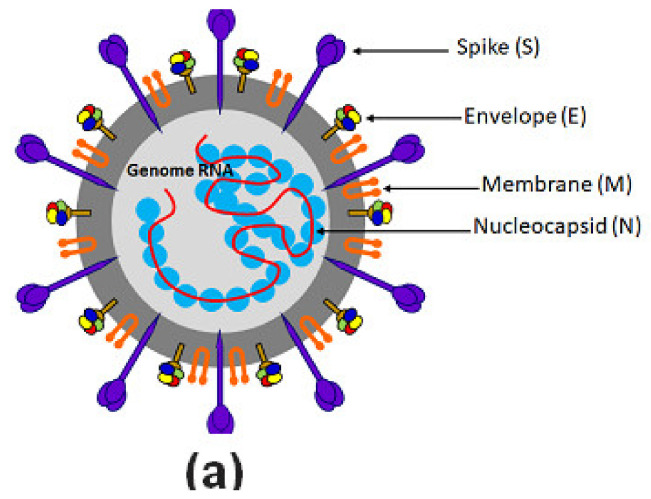
SARS-CoV-2 structure representation. (**a**) Schematic model for location of the main surface proteins of the new coronavirus. (**b**) Genome structure organization of open reading frames (ORFs) for structural, accessory, and non-structural proteins and their relative genome positions.

**Figure 2 vaccines-08-00692-f002:**
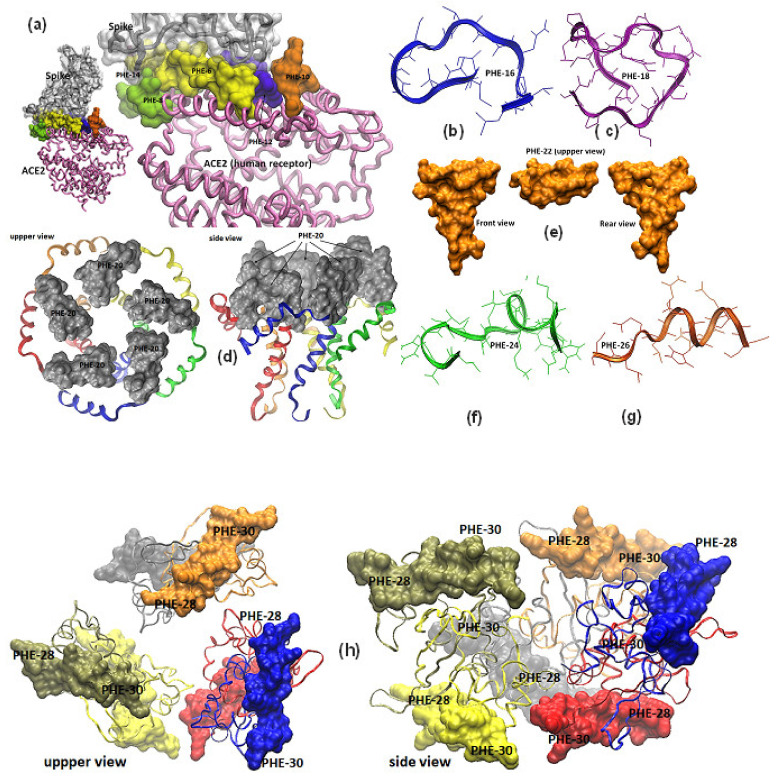
Molecular models of SARS-CoV-2 surface proteins and relative location of designed epitope-peptide targets. (**a**) Representation of the SARS-CoV-2 spike receptor-binding domain complexed with angiotensin converting enzyme 2 (ACE2), protein data bank (PDB) file code 6m0j [7], epitope-peptides PHE-2, PHE-4, PHE-6, PHE,8, PHE-10, PHE,12 and PHE14 are highlighted in the interface of the 3D structure. PepFold molecular prediction models for epitope-peptides PHE-16 and PHE-18 from ORF3a are shown in blue and purple ribbons in (**b**,**c**) Pentamer envelope small membrane protein-E (PDB code 5 × 29) of a formed ion channel resolved by NMR experiments, in which the epitope-peptide PHE-20 is displayed as a gray surface in all protein chains [9]. (**d**) The M protein (membrane or matrix protein) is a triple-spanning membrane arranged, predictive M-based epitope-peptidePHE-22 3D structure was analyzed by PepFold and represented as a yellow mustard solid surface. (**e**) Modified epitope-peptides PHE-22 and PHE-24 based on the expression product of ORF7a were molecular modelized by the PepFold predictive 3D structure algorithms and herein represented as green and orange ribbons in (**f**,**g**). (**h**) Dimers associated with a hexamer complex 3D structure (PDB codes 6m3m and 6wkp) were resolved by X-ray crystallography experiments for the nucleocapsid RNA binding domain of N protein and herein is represented as colorful ribbons in which peptides PHE-28 and PHE-30 are highlighted as solid surfaces.

**Figure 3 vaccines-08-00692-f003:**
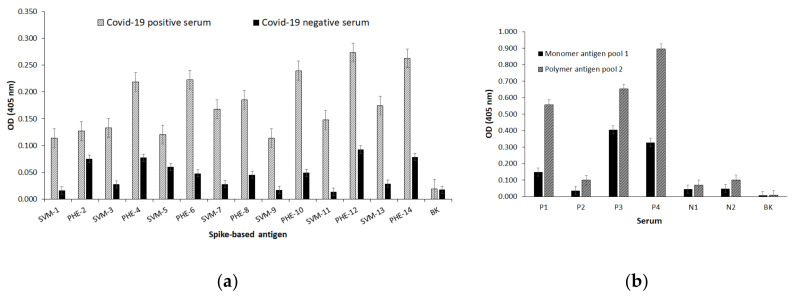
Human serum from two PCR diagnosed patients´ immuno-reactivity for synthesized epitope-peptides based on the SARS-CoV-2 spike protein. (**a**) Molecular recognition of S-based epitope-peptides presented as monomer (SMV) and polymer (PHE) forms. (**b**) Serum from four patient’s PCR positive diagnosed (P) strongly reacts with equimolar pooled S-based polymer epitopes while two negative diagnosed (N) had no reactivity. (**c**) The same panel of serum from four patients PCR positive diagnosed (P) strongly reacts with individual E, M, N, ORF3a, ORF3b, ORF7a, 7b-based epitopes presented as monomer (SVM) and polymer (PHE) forms while two negative diagnosed (N) had no reactivity. Antigen controls were coded as C31 and C33 representing two influenza virus A H1N1 epitope-peptides. CC represents a reagents control and BK the blank.

**Figure 4 vaccines-08-00692-f004:**
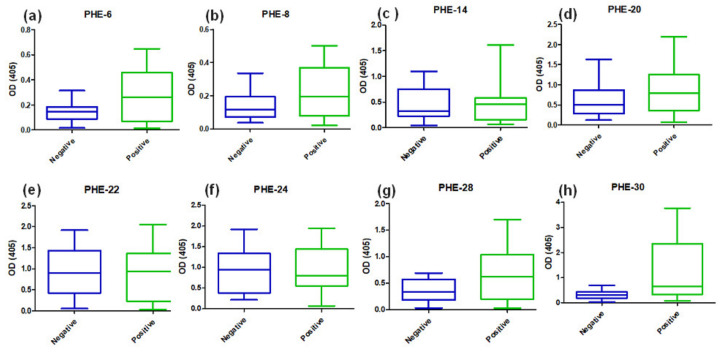
Serological reaction of COVID-19 positive and negative PCR diagnosed subjects for SARS-CoV-2-designed synthetic epitope-peptides. A panel of 20 human serum samples obtained from PCR positive COVID-19 diagnosed patients, and 20 human serum samples from negative diagnosed subjects, were tested for their immuno-reactivity for epitope-peptides representing the whole panel of SARS-CoV-2 target proteins S, E, M, N, 3a and 7a of this work. A differential epitope molecular recognition is revealed in these multiple tests. Tested epitope-peptides were (**a**) PHE-6 from ORF2, (**b**) PHE-8 from ORF2, (**c**) PHE-14 from ORF2, (**d**) PHE-20 from ORF4, (**e**) PHE-22 from ORF5, (**f**) PHE-24 from ORF7A, (**g**) PHE-28 fromORF9 and (**h**) PHE-30 from OFR9.

**Figure 5 vaccines-08-00692-f005:**
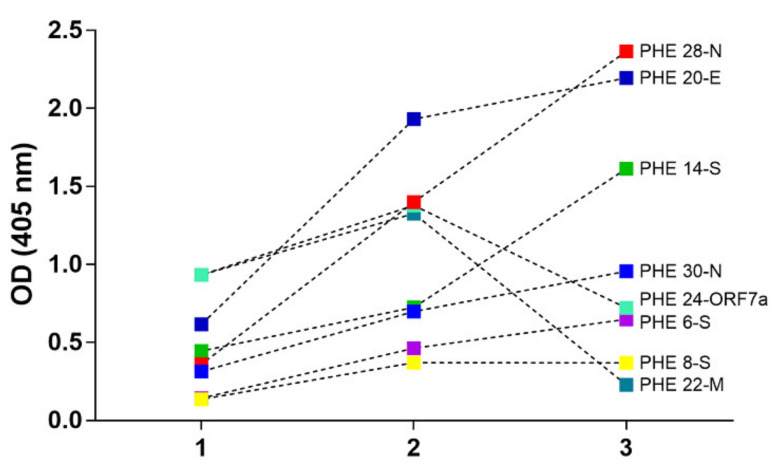
Evidence of a long-lasting humoral immunity for specific SARS-CoV-2 epitope-peptides. Patients−some cases of study became evident. A representative case study reveals an interesting immune-reactivity pattern by all different tested epitope-peptides representative of gene expression products S, E, M, N, 3a and 7a, respectively, in a follow-up between 1 month (lane 2) and 5 months (lane 3) after becoming infected with the SARS-Cov-2. Lane 1 represents the average of negative controls.

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
