# Peer review of "COVID-19 Infection Detection and Prevention by SARS-CoV-2 Active Antigens: A Synthetic Vaccine Approach"

_vaccines, 2020, doi:10.3390/vaccines8040692_

Round 1

Reviewer 1 Report

In this paper, the authors predicted the fifteen SARS-CoV-2 antigen peptides by using bioinformatic tools. They synthesized them and tested their immunoreactivities by using the plasmas from COVID-19 positive patients. Overall, this paper provides a rationale design scheme necessary for developing peptides as synthetic vaccines. Here are some suggestions for authors.

  1. There are so many sentences that are grammatically incorrect and very hard to understand. English editing by a native speaker is absolutely required for entire manuscripts.
  2. In addition to their structural information shown in figure2, full amino acid sequences of all the fifteen peptides need to be revealed for a more clear understanding of the potential functions as vaccine candidates.
  3. Covid-19 is written as COVID-19 in most of the literature these days.
  4. In the introduction, general information about the genetic map of SARS-CoV2 needs to be shortened. Instead, the authors need to add a paragraph on how to find the potential antigenic peptides by using bioinformatic tools.
  5. In line 181, there is no need to add the full mailing address of NCBI.
  6. The authors need to define “LB epitope” in line 184. Is this “linear B cell epitope”?
  7. The link https://prosper.erc.monash.edu.au/ shown in line 197 is not working.
  8. The link https://bioserv.rpbs.univ-paris-302diderot.fr/services/PEP-FOLD/ shown in line 302 is not working either.
  9. The nomenclature of peptides is very confusing. Are SVM and PHE monomeric and polymeric forms of peptides? Are SVM-1 and PHE-2 the same peptides and differ only in monomeric and polymeric states? Why do they use the different numbers in the same peptides?
  10. Why are the polymeric forms of peptides more immunogenic to patients plasma than the monomeric forms of peptides?

Author Response

Reviewer 1 comments and suggestions:

Comment: In this paper, the authors predicted the fifteen SARS-CoV-2 antigen peptides by using bioinformatic tools. They synthesized them and tested their immunoreactivities by using the plasmas from COVID-19 positive patients. Overall, this paper provides a rationale design scheme necessary for developing peptides as synthetic vaccines. Here are some suggestions for authors.

Authors´ answer: Thank you for your encouraging and positive point of view regarding our work

Specific comments:

  1. There are so many sentences that are grammatically incorrect and very hard to understand. English editing by a native speaker is absolutely required for entire manuscripts.

Authors´ answer: In agreement, the whole manuscript text was carefully reviewed and adjusted by a native English speaker.

  1. In addition to their structural information shown in figure2, full amino acid sequences of all the fifteen peptides need to be revealed for a moreclear understanding of the potential functions as vaccine candidates.

Authors´ answer: Thank you for this comment. Bearing in mind that most active sequences presented in the manuscript are being assessed for their immunological properties and giving their relevance for a potential vaccine candidate composition we decide to intellectually protect them by a Provisional patent application filed on the USPTO as denoted in the Patents section of the manuscript. However, we have slightly modified some statements in the manuscript especially in Fig.2 in order to provide a clearer explanation for a better understanding this work message and findings.

  1. Covid-19 is written as COVID-19 in most of the literature these days.

Authors´ answer: In agreement, this acronym for the disease was adjusted in the text as kindly suggested.

  1. In the introduction, general information about the genetic map of SARS-CoV2 needs to be shortened. Instead, the authors need to add a paragraph on how to find the potential antigenic peptides by using bioinformatic tools.

Authors´ answer: In agreement, adjusts were performed on this section of the manuscript.

  1. In line 181, there is no need to add the full mailing address of NCBI.

Authors´ answer: Proposed change was adjusted accordingly and highlighted in the referenced line.

  1. The authors need to define “LB epitope” in line 184. Is this “linear B cell epitope”?

Authors´ answer: LB epitope was properly defined as requested and highlighted in the referenced line.

  1. The link https://prosper.erc.monash.edu.au/ shown in line 197 is not working.

Authors´ answer: In agreement, an up dated version of this URL is now provided in the revised version of the manuscript.

  1. The link https://bioserv.rpbs.univ-paris-302diderot.fr/services/PEP-FOLD/ shown in line 302 is not working either.

Authors´ answer: As requested in the above comment, the text provide now an updated version for this URL.

  1. The nomenclature of peptides is very confusing. Are SVM and PHE monomeric and polymeric forms of peptides? Are SVM-1 and PHE-2 the same peptides and differ only in monomeric and polymeric states? Why do they use the different numbers in the same peptides?

Authors´ answer: In agreement, some new statements and explanations in order to clearly define the proposed nomenclature and its meaning was introduced in the revised version of the manuscript.

  1. Why are the polymeric forms of peptides more immunogenic to patients plasma than the monomeric forms of peptides?

Authors´ answer: In line with this relevant comment, some new statements were also added to different sections of the manuscript in order to make the discussion and conclusions stronger. These new statements were highlighted in the text for any further revision.

Reviewer 2 Report

The manuscript is interesting and well written. It is acceptable for publication in the current form

The strengths and weaknesses of this paper is based upon the topic discussed, the structure of the paper, the adequate description of the results.

Author Response

Reviewer 2 comments and suggestions:

Comment: The manuscript is interesting and well written. It is acceptable for publication in the current form.

The strengths and weaknesses of this paper is based upon the topic discussed, the structure of the paper, the adequate description of the results.

Authors´ answer: Thank you for your encouraging and positive comments regarding our work

Round 2

Reviewer 1 Report

All my concerns were addressed.

Thanks